# Effects of Host Plants and Their Infection Status on Acquisition and Inoculation of A Plant Virus by Its Hemipteran Vector

**DOI:** 10.3390/pathogens12091119

**Published:** 2023-09-01

**Authors:** Saurabh Gautam, Kiran R. Gadhave, James W. Buck, Bhabesh Dutta, Timothy Coolong, Scott Adkins, Alvin M. Simmons, Rajagopalbabu Srinivasan

**Affiliations:** 1Department of Entomology, University of Georgia, 1109 Experiment Street, Griffin, GA 30223, USA; sg37721@uga.edu; 2Texas A&M AgriLife Research, 6500 W Amarillo Blvd, Amarillo, TX 79106, USA; kiran.gadhave@ag.tamu.edu; 3Department of Plant Pathology, University of Georgia, 1109 Experiment St., Griffin, GA 30223, USA; jwbuck@uga.edu; 4Department of Plant Pathology, University of Georgia, 3250 Rainwater Road, Tifton, GA 31793, USA; bhabesh@uga.edu; 5Department of Horticulture, University of Georgia, 1111 Miller Plant Sciences, 120 Carlton Street, Athens, GA 30602, USA; tcoolong@uga.edu; 6USDA-ARS, U.S., Horticultural Research Laboratory, Fort Pierce, FL 34945, USA; scott.adkins@usda.gov; 7USDA-ARS, U.S., Vegetable Laboratory, Charleston, SC 29414, USA; alvin.simmons@usda.gov

**Keywords:** transmission, *B. tabaci*, begomovirus, crinivirus, virus accumulation

## Abstract

Whitefly, *Bemisia tabaci* Gennadius (B cryptic species), transmits cucurbit leaf crumple virus (CuLCrV) in a persistent fashion. CuLCrV affects several crops such as squash and snap bean in the southeastern United States. CuLCrV is often found as a mixed infection with whitefly transmitted criniviruses, such as cucurbit yellow stunting disorder virus (CYSDV) in hosts such as squash, or as a single infection in hosts such as snap bean. The implications of different host plants (inoculum sources) with varying infection status on CuLCrV transmission/epidemics is not clear. This study conducted a series of whitefly mediated CuLCrV transmission experiments. In the first experiment, three plants species: squash, snap bean, and tobacco were inoculated by whiteflies feeding on field-collected mixed-infected squash plants. In the second experiment, three plant species, namely squash, snap bean, and tobacco with varying infection status (squash infected with CuLCrV and CYSDV and snap bean and tobacco infected with CuLCrV), were used as inoculum sources. In the third experiment, squash plants with differential CuLCrV accumulation levels and infection status (either singly infected with CuLCrV or mixed infected with CuLCrV and CYSDV) were used as inoculum sources. Irrespective of plant species and its infection status, CuLCrV accumulation in whiteflies was dependent upon the CuLCrV accumulation in the inoculum source plants. Furthermore, differential CuLCrV accumulation in whiteflies resulted in differential transmission, CuLCrV accumulation, and disease phenotype in the recipient squash plants. Overall, results demonstrate that whitefly mediated CuLCrV transmission between host plants follows a virus density dependent phenomenon with implications for epidemics.

## 1. Introduction

Successful survival of plant pathogens in the natural ecosystem depends on the availability of susceptible hosts [1]. A host is a plant that a pathogen can infect and within which the pathogen can replicate. Many pathogens have a broad host range, which ensures the pathogen’s survival within the ecosystem [2,3]. Plant viruses, for the most part, depend upon vectors, particularly insects for transmission [4]. Therefore, in the case of vector-transmitted plant viruses, an ideal host that supports virus multiplication should allow virus acquisition by vector/s for successful transmission [5]. Hence, in the case of vector-borne plant viruses, understanding the host range within the context of host suitability for vector transmission is central to understanding virus pathogenicity and epidemics. In the last decade, the annual occurrence of the whitefly *Bemisia tabaci* Gennadius B (otherwise referred to as B biotype/B mitotype/MEAM1) [6,7] transmitted cucurbit leaf crumple virus (family *Geminiviridae*, genus *Begomovirus*) is increasingly becoming important in squash (*Cucurbita pepo* L.) and snap bean (*Phaseolus vulgaris* L.) production in the southeastern United States [8,9,10,11]. CuLCrV was first detected in the southeastern U.S., in 2006, in Florida [12], and in 2010, it was detected in snap bean in Georgia [13]. Recently, CuLCrV was detected in cucurbits in South Carolina, demonstrating its expanding geographical range [14]. Although the host range of cucurbit leaf crumple virus (CuLCrV) and CuLCrV-mediated direct and indirect effects on vector fitness were studied previously [9], the ability of *B. tabaci* B to acquire and inoculate CuLCrV from these hosts remains largely unknown.

Polyphagous vectors, such as *B. tabaci* B, moving through a habitat may encounter multiple host plants and can acquire viruses during the process, resulting in single- or mixed infection in subsequently encountered susceptible host plants [8]. Mixed infection is ubiquitous in nature [15,16,17], and many co-infecting viruses interact synergistically [18]. Several agriculturally important plant viruses produce enhanced disease phenotypes in a mixed infection status [19,20,21,22,23], resulting in altered interactions with insect vectors with implications for virus transmission [20,24,25,26,27,28]. In squash, CuLCrV mostly occurs in mixed infection with other *B. tabaci* B-transmitted viruses, such as cucurbit chlorotic yellows virus (CCYV, family *Closteroviridae*, genus Crinivirus) and/or cucurbit yellow stunting disorder virus (CYSDV, genus Crinivirus) [9,29]. Compared with single infection, mixed-infected squash often produces a severe disease phenotype [9]. On mixed-infected squash, CuLCrV infection symptoms (crumpling) are more pronounced in upper leaves and CYSDV-specific symptoms (interveinal chlorosis) are prominent in older leaves.

Cucurbit leaf crumple virus and criniviruses (CCYV and CYSDV) differ in their interactions with *B. tabaci*. CuLCrV is transmitted by *B. tabaci* in a persistent and circular manner. Once acquired by the whitefly, CuLCrV can remain associated with it for the rest of the life span [30]. Criniviruses are semipersistent viruses, and once acquired by whiteflies feeding on infected plants; whiteflies can remain viruliferous for up to nine days [31,32]. Unlike CCYV, which was documented in 2019, CYSDV has been known to occur in the region for more than a decade [8,33]. Both CuLCrV and CYSDV can infect many plant species in the family *Cucurbitaceae*, and certain cultivars of the common or snap bean (*Phaseolus vulgaris* L.) [34,35,36,37]. Squash and snap bean are two commercially important crops in Georgia, where in 2021, growers planted ~4382 and 9873 acres of squash and snap bean, respectively, with a combined farmgate value of more than USD 74 million [38]. Also, both share a growing season in Georgia. During severe epidemics, field infection of CuLCrV in squash and snap bean could reach up to 100% in susceptible cultivars [11]. In squash, CuLCrV is often found to be mixed infected with CYSDV [13,35,39]. The frequent mixed infection of CuLCrV with CYSDV raises two important epidemiological questions. First, how does the acquisition of one virus (CuLCrV) and two (CuLCrV and CYSDV) viruses by *B. tabaci* affect its ability to transmit CuLCrV? Second, how the inoculum source infection status (single: CuLCrV and mixed infection: CuLCrV and CYSDV) could affect the CuLCrV spread/epidemic?

To answer the above questions, the current study used field-collected, mixed-infected squash with CuLCrV and CYSDV as an initial inoculum source for CuLCrV, and conducted a series of *B. tabaci* B-mediated transmission experiments (Figure 1): (i) CuLCrV transmission from mixed-infected squash to tobacco (experimental host), snap bean, and squash; (ii) CuLCrV transmission from mixed- (squash) versus singly infected (tobacco and snap bean) plants to squash; and (iii) CuLCrV transmission from mixed- versus singly infected squash to squash (Figure 1). In doing so, this study also quantitated the amount of CuLCrV acquired by the whiteflies feeding on mixed- versus singly infected plants belonging to different families. Also, this study estimated CuLCrV accumulation in squash plants inoculated by whiteflies, which acquired CuLCrV from different inoculum sources (mixed- and singly infected squash and singly infected tobacco and snap bean).

## 2. Materials and Methods

### 2.1. Plants and Insects

In the transmission studies, three different plant species, squash (*Cucurbita pepo* cv. “Gold Star, F1 hybrid”), snap bean *P. vulgaris* cv. “Provider” and tobacco (*Nicotiana tabacum* cv. “L326”), were used. Yellow summer squash and snap bean seeds were procured from Johnny’s Selected Seeds (Winslow, ME, USA), and tobacco seeds were obtained from UGA extension services. Throughout the study, two-week-old squash, four-week-old tobacco, and four-week-old snap bean seedlings were used for transmission experiments. Plants were at the four-leaf stage in all experiments. Plants were maintained in a greenhouse (25–30 °C with a 14 h L:10 h D photoperiod) in insect-proof cages (Megaview Science Co., Taichung, Taiwan) at five plants per cage. The whiteflies (*B. tabaci* ‘B biotype’) used in the present study were first collected in 2009 in Tifton, Georgia from infested cotton (*Gossypium hirsutum* L.) fields. Since then, whiteflies were reared on cotton plants in whitefly proof cages in the greenhouse maintained at the above-stated conditions [40]. The purity of the colony was periodically confirmed (once every few months) by partially sequencing the mitochondrial cytochrome oxidase I (COI) gene [41].

### 2.2. Virus Maintenance

The initial inoculum source of CuLCrV and CYSDV was obtained using the procedure described earlier by Gautam et al. 2020 [9]. Briefly, in September 2017, fifteen whitefly infested squash plants showing symptoms, such as crumpling and yellowing, were collected from a research plot in Tifton, GA. Symptomatic leaves were excised and surface sterilized using a six-step surface sterilization. The leaves were first washed in autoclaved distilled water, followed by 1 min rinsing in 1% bleach, followed by a 1 min wash in 70% ethanol, and finally, three rinses with sterile distilled water to remove the sterilizing agents. In total, 100 mg of surface-sterilized leaf tissues were used for DNA/RNA extraction. Total genomic DNA was extracted with the GeneJET Plant Genomic Purification Kit (ThermoFisher Scientific, Waltham, MA, USA) following the manufacturer’s protocol. Total RNA was extracted using the RNeasy Plant Mini Kit (Qiagen, Valencia, CA, USA) following the manufacturer’s guidelines and subjected to cDNA synthesis. The presence of CuLCrV in extracted DNA and CYSDV in generated cDNA was confirmed using the primers and conditions described earlier [9]. All fifteen samples tested positive for both CuLCrV and CYSDV. The inoculum was maintained in the greenhouse through repeated whitefly mediated inoculation in whitefly proof cages by placing two-week-old, non-infected squash plants with the infected plants.

### 2.3. Transmission Experiment One (E1)

*Bemisia tabaci* B mediated-CuLCrV transmission was conducted using field-collected mixed-(CuLCrV and CYSDV) infected squash as the inoculum source, and squash, tobacco and snap bean as recipients. Around one thousand whiteflies without CuLCrV (upto 48 h old) feeding on cotton were given a 48 h acquisition access period (AAP) on mixed-infected squash. Using a clip cage, whiteflies (~100/plant) that acquired both viruses were attached to the first true leaf of tobacco, snap bean, or squash plants (Gautam et al. 2020) for an inoculation access period (IAP) of five days. Clip cages were made of foam and each cage comprised two hollow foam rings (inner diameter of 2.5 cm, and a height of 1.5 cm) with mesh covering one side of each ring for visibility and ventilation. Using pins, as shown in Appendix A, plant leaf tissues were sandwiched between the rings of clip cages containing insects. After four weeks, the total DNA/RNA from 100 mg of young leaf tissue was extracted and subjected to PCR analysis for CuLCrV and CYSDV, as described above. Each treatment had 10 replicates (one clip cage/plant), and the experiment was conducted three times (n = 30). For each experiment (n = 3), different inoculum sources (field collected mixed-infected plants) were used.

CuLCrV copy numbers in infected plants were estimated using the qPCR protocol described earlier by Gautam et al. 2020 [9]. Briefly, 2X GoTaq qPCR Master Mix was combined with forward and reverse primers (CuLCrV-QF and CuLCrV-QR final concentration of 0.5 μM), 10 ng DNA, and nuclease-free water for a final reaction volume of 25 μL. Cycling parameters were as follows: 95 °C for 2 min; 40 cycles of 95 °C for 1 min, 63 °C for 15 s, and 72 °C for 20 s. Upon completion of the run, melting curve analysis was performed to confirm the specificity of the primer pairs. Each sample was tested in duplicate, and the absolute number of copies in the samples was quantitated using the standard curve protocol described by Legarrea et al. [40]. Generated singly and mixed-infected plants were used as inoculum sources in transmission experiment two, as described below.

### 2.4. Transmission Experiment Two (E2)

*Bemisia tabaci* B-mediated CuLCrV transmission was conducted using singly (CuLCrV: tobacco and snap bean) and mixed- (CuLCrV and CYSDV: squash) infected inoculum sources and squash plants as recipients.

*Whitefly acquisition*: Whiteflies with or without viruses were obtained by allowing whiteflies to feed on non-infected, singly or mixed-infected inoculum sources for a 48 h AAP. Following AAP, whiteflies were attached to cotton plants using clip cages (100/clip cage). After 48 h on cotton, twenty whiteflies were randomly collected, and surface sterilized using the six-step protocol described above. For CuLCrV, total DNA was extracted from individual whiteflies using Chelex resin, InstaGene Matrix (Bio-Rad, Hercules, CA, USA) and subjected to PCR analysis, as described above. Percent infection and CuLCrV accumulation in whiteflies were estimated using endpoint PCR and qPCR programs, as described above. Whiteflies with 48 h AAP on non-infected squash plants served as control. The experiment was conducted three times (n = 60 for each treatment).

*Whitefly inoculation*: Using clip cages, whiteflies with or without the virus (100 adults/plant) were attached to the first true leaf of two-week-old squash and given an IAP of five days. Four weeks after the five-day IAP, total DNA/RNA from 100 mg surface-sterilized young leaf tissue was extracted and subjected to PCR analysis, as described above. Percent CuLCrV and/or CYSDV infection in squash was measured using endpoint PCR protocol, as described above, and CuLCrV copy numbers in infected squash were estimated using the protocol, as described above. Each treatment had 10 replicates and the experiment was conducted three times (n = 30). Generated ingly (squash infected from tobacco and snap bean, hereafter referred as ST and SB, respectively) and mixed-infected (squash infected from squash, hereafter referred SM)) squash plants were used as inoculum sources in transmission experiment three, as described below.

### 2.5. Transmission Experiment Three (E3)

*Bemisia tabaci* B mediated CuLCrV transmission was conducted using CuLCrV-infected squash ((SB or ST) obtained from singly infected snap bean or tobacco) and CuLCrV and CYSDV-infected squash ((SM) obtained from mixed-infected squash) as inoculum sources and non-infected squash plants as recipients. Whiteflies with or without viruses were generated by providing whiteflies with a 48 h AAP on non-infected, singly (SB or ST), or mixed infected squash (SM). Hereafter, whiteflies feeding on SB, ST, and SM will be referred to as WSB, WST, and WSM, respectively. CuLCrV transmission from singly/mixed-infected squash was carried out using the protocol described above. Percent infection and CuLCrV accumulation in whiteflies and plants were estimated using the protocol described above. Each treatment had 10 replicates and the experiment was conducted three times (n = 30).

### 2.6. Statistical Analysis

Whitefly and plant percentage infection data (infected versus. non-infected) were evaluated using binary logistic regression. CuLCrV accumulation data in plants were analyzed using a linear-mixed model. During analysis, replications and repeats were considered as random effects and treatments were considered fixed effects. Means separation was performed using Tukey’s HSD post hoc test. Data analyses were performed in R version 3.4.2 [42].

## 3. Results

### 3.1. Transmission Experiment One (E1)

Inoculated tobacco and snap bean plants were infected only with CuLCrV and not with CYSDV. However, inoculated squash plants were infected with both CuLCrV and CYSDV. Significantly higher infection percentages were observed in squash (93%) compared with tobacco (68%) and snap bean plants (62%) (χ^2^ = 14.33; df = 2, 87; *p* < 0.01; Figure 2A). CuLCrV infection symptoms were observed three to five weeks post-inoculation. In infected squash plants, crumpling on younger leaves was accompanied by interveinal chlorosis on older leaves (Figure 3A). Symptoms on tobacco plants included thickened and leathery young leaves (Figure 3B), and severe crumpling was observed on snap bean younger leaves (Figure 3C). CuLCrV accumulation differed significantly between the plant species (*F* = 10.43; df = 2, 68; *p* < 0.01). The highest CuLCrV accumulation was observed in mixed-infected squash and the lowest accumulation was documented in snap bean plants (Figure 2B).

### 3.2. Transmission Experiment Two (E2)

The percentage of whiteflies that acquired CuLCrV differed significantly depending upon the acquisition host plant species (χ^2^ = 18.41, df = 2, 177; *p* < 0.01). The highest percent was observed for whiteflies feeding on mixed-infected squash (95%), followed by tobacco (83%) and snap bean (64%) (Figure 4A). CuLCrV accumulation in whiteflies differed significantly depending upon the host plant species (*F* = 62.304, df = 2, 144, *p* < 0.01). CuLCrV accumulation in whiteflies feeding on mixed-infected squash was significantly higher than that of whiteflies feeding on tobacco or snap bean (Figure 4B). Within single infections, whiteflies feeding on tobacco acquired significantly more CuLCrV than whiteflies feeding on snap bean (Figure 4B).

Percent infection and accumulation differed in infected squash plants depending on the CuLCrV inoculum source. The highest percent infection was observed in squash (SM) inoculated by whiteflies post-acquisition on mixed-infected squash (84%) and the lowest percent infection was observed in squash (SB) inoculated by whiteflies post-acquisition on infected snap bean (40%) (χ^2^ = 129.7; df = 2, 87; *p* < 0.01) (Figure 4C). CuLCrV accumulation in infected squash was dependent on the CuLCrV inoculum source plants (*F* = 628.6; df = 2, 72; *p* < 0.01). There was no significant difference in the CuLCrV accumulation in squash (SM and ST) inoculated by whiteflies post-acquisition on mixed-infected squash and singly infected tobacco. Squash (SB) inoculated by whiteflies post-acquisition on infected snap bean plants accumulated significantly lower amounts of CuLCrV than SM and ST squash inoculated by whiteflies post-acquisition on mixed-infected squash or singly infected tobacco plants (Figure 4D). Symptom development in squash varied depending upon the CuLCrV inoculum source (Figure 3). Mixed-infected squash (SM) had severe crumpling (Figure 3D). Squash (ST) inoculated by whiteflies post-acquisition on infected tobacco that had characteristic CuLCrV symptoms (curled and crumpled young leaves) (Figure 3E). Mild symptoms (thickened leaves) developed on squash (SB) inoculated by whiteflies post-acquisition on infected snap bean plants (Figure 3F).

### 3.3. Transmission Experiment Three (E3)

The percentage of whiteflies that acquired CuLCrV was dependent on the CuLCrV accumulation levels in squash (SB, ST, or SM) (χ^2^ = 61.81; df = 2, 87; *p* < 0.01) (Figure 5A). The highest percentage (95%) of whiteflies positive for CuLCrV (WSM) was observed while feeding on mixed-infected squash plants (SM). Within single infections, 77% of CuLCrV positive whiteflies (WST) were observed while feeding on singly infected plants, ST, and the lowest percentage (40%) of CuLCrV positive whiteflies (WSB) was observed while feeding on singly infected plants, SB.

CuLCrV-DNA accumulation did not differ between WSM and WST whiteflies (*F* = 73.8; df = 2, 124; *p* <0.01) (Figure 5B). However, WSM and WST whiteflies accumulated significantly more CuLCrV DNA than WSB whiteflies.

Percent infection and CuLCrV accumulation in infected squash were dependent up-on the CuLCrV inoculum source (SB, ST, or SM). The highest percent infection was observed in squash plants (85%) inoculated by whiteflies (WSM) post-acquisition on mixed-infected squash plants (SM). Within single infection sources, the percent infection in squash plants (80%) inoculated by WST whiteflies was higher than the squash plants (20%) inoculated by WSB whiteflies (χ^2^ = 60.92; df = 2, 6; *p* < 0.01) (Figure 5C).

CuLCrV-DNA accumulation did not differ between mixed- and singly infected squash plants inoculated by WSM and WST whiteflies post-acquisition on mixed-infected (SM) and singly infected squash (ST) (Figure 5B). However, these plants accumulated significantly higher CuLCrV DNA than squash inoculated by WSB whiteflies post-acquisition on singly infected squash plants (SB) (*F* = 478.42; df = 2, 57; *p* < 0.01) (Figure 5D).

Mixed-infected squash inoculated by WSM whiteflies had severely crumpled leaves (Figure 3G). Singly infected squash inoculated by WST whiteflies had characteristic CuLCrV symptoms (curled and crumpled young leaves) (Figure 3H). Mild symptoms (slower growth and stunting) developed on squash plants inoculated by WSB whiteflies (Figure 3I).

## 4. Discussion

Insect-mediated plant virus transmission into new hosts, and multiplication within new hosts are two key components of virus fitness, which ensure survival within the ecosystem [43]. Understanding the insect vector-mediated transmission cycle within in an agroecosystem is fundamental to comprehending virus epidemiology and developing management plans. The current study examined the transmission of CuLCrV by *B. tabaci* B feeding on CuLCrV-infected plants with differential infection status and varying levels of CuLCrV accumulation. Irrespective of CuLCrV-inoculum source plant species and their infection status, the percent infection and CuLCrV accumulation in whiteflies were proportional to the CuLCrV accumulation in the inoculum source (Appendix A). Furthermore, disease phenotype development and percent infection in inoculated squash plants were dependent on the amount of CuLCrV present in the inoculating whiteflies. Overall, results from the current study provide experimental evidence that *B. tabaci* B-mediated CuLCrV epidemics in squash largely depend on the CuLCrV accumulation in the inoculum source. Knowledge about the likely level of CuLCrV in reservoir plants may be helpful in improving strategies for controlling CuLCrV epidemics in the fields.

Whiteflies were able to acquire CuLCrV from both mixed- (squash) and singly infected plants (tobacco, snap bean, squash), confirming that CuLCrV accumulation in plants was above the threshold that would limit its acquisition. Depending on the host suitability, *B. tabaci* B is reported to feed differentially on different host plants [44]. Differential feeding can affect the virus transmission by whiteflies [45]. Therefore, results obtained in the second transmission experiment that assessed CuLCrV transmission from mixed- (squash) versus singly infected (tobacco and snap bean) plants to squash might be the result of differential feeding of *B. tabaci* B on squash, tobacco, and snap bean along with differential CuLCrV accumulation in the three hosts. However, in the third transmission experiment through squash-to-squash transmission assays (CuLCrV transmission from mixed- versus singly infected squash), the host plant variable was removed from the experiment. Results from both the second and third transmission experiments demonstrated that CuLCrV accumulation in whiteflies was dependent on the virus accumulation in the host plants. Similar patterns of begomovirus accumulations in whiteflies that are dependent on the host plants have been reported earlier for tomato yellow leaf curl virus (TYLCV) [46] and sida golden mosaic virus (SiGMV) [5]. Furthermore, studies with solutions of purified virions of TYLCV have also proved that TYLCV accumulation in whiteflies is proportional to the virus load in the inoculum source [47]. Begomoviruses use receptor-mediated endocytosis to enter and accumulate within whiteflies tissues (midgut and salivary gland). In all the studies cited above, whiteflies were given sufficient AAP (48–72 h). Theoretically, TYLCV, SiGMV, and CuLCrV should have reached the maximum accumulation threshold within the provided AAP.

In the third transmission experiment, within singly infected plants, squash inoculated by whiteflies (WSB) feeding on squash (SB) that was inoculated by whiteflies post-acquisition on infected snap bean had mild symptoms, which disappeared after eight weeks of infection (data not shown). It is possible that reduced multiplication of CuLCrV in snap bean might be due to selection for a virus isolate/variant that is less capable of multiplication in snap bean. Recovery phenotypes observed in certain begomovirus-infected plants, coupled with a reduction in virus accumulation and symptom severity, could have been aided by host defense responses [48,49,50]. Begomoviruses are reported to trigger the host’s defense systems, which can lead to recovery from the infection [51,52,53,54], but begomoviruses also can suppress the host’s defense system [55,56]. Therefore, infection, at least partially, depends on the ability of begomoviruses to suppress the host’s defense system. In the case of aphid-transmitted potyviruses, a couple of studies measured the inoculum threshold of virus particles required to induce infection [57,58]. However, studies quantifying inoculum thresholds for begomovirus particles to induce and/or sustain infection are lacking. Results from the current study and studies with potyviruses imply that in the third transmission experiment, whiteflies feeding on singly infected squash (E2) inoculated by whiteflies post-acquisition from singly infected snap bean (E1) did not inoculate enough CuLCrV particles into squash for the virus to overcome the host’s defense response. However, in the current study, CuLCrV accumulation is estimated in whole insects; the amount of virus accumulated in the primary salivary glands and the actual amount of virus inoculated by whiteflies carrying CuLCrV remains unknown.

Under field conditions, effects of reduced CuLCrV accumulation in inoculum sources and subsequent reduced acquisition of CuLCrV in whiteflies could be enhanced by the presence of mixed infection (CCYV and/or CYSDV) in the inoculum source, leading to an irreversible (unrecoverable) phenotype. Interactions between plant viruses in a mixed infection can be synergistic, antagonistic, and neutral. Synergistic interactions resulted in increased disease severity compared with single virus infection [59,60]. In squash, CuLCrV often occurs as a mixed infection with CCYV and/or CYSDV [9,29]. Mixed infection can occur via multiple scenarios: by whiteflies that acquire virions from mixed-infected plants; sequentially acquire from singly infected plants or by multiple whitefly individuals that acquir different viruses. Currently, it is not known if the reduced CuLCrV inoculation by whiteflies in squash in some scenarios would be compensated by the presence of CCYV and/or CYSDV to kick start the irreversible CuLCrV infection/phenotype. Vector-borne pathogens are known to influence the host’s phenotype in ways that stimulate their transmission by the vector [61,62]. Previously, studies conducted using the same population of *B. tabaci* B under similar conditions indicated that the interactions between the begomovirus, host, and vector are pathosystem specific. For instance, studies with TYLCV-infected tomato demonstrated that *B. tabaci* B that acquired no virus were attracted towards susceptible genotypes with higher TYLCV accumulation, and they accumulated higher TYLCV compared with whiteflies feeding on resistant hosts with reduced TYLCV accumulation [40]. Also, *B. tabaci* B developmental time decreased significantly on TYLCV-infected susceptible tomato plants compared with non-infected plants. In contrast, in another study, *B. tabaci* B that acquired no virus, avoided settling on squash infected with CuLCrV, and whitefly development on CuLCrV-infected squash did not result in any fitness benefits [9]. CuLCrV can infect multiple host plants belonging to different families, more research on CuLrV-host-vector interactions using different host plants is warranted to fully comprehend the CuLCrV disease cycle in squash and snap bean agro-ecosystems.

## 5. Conclusions

CuLCrV was first reported from the Imperial Valley of California in 1988 [63], and did not cause serious production issues until it was introduced into the Southeastern United States around 2006 [12]. Processes behind virus disease emergence are complex, and predicting what will trigger the next epidemic is difficult. Given that viruses are reported not to cause apparent disease in wild plants because they cannot afford to kill their reservoir host [17,64], there are three possible scenarios for annual epidemics of CuLCrV. First, there could be wild host plants that accumulate high levels of CuLCrV and serve as long-term reservoirs and efficient inoculum sources for CuLCrV. Second, the large populations of virus-laden whiteflies (as seen in the southeast during summer and fall) with low amounts of CuLCrV might be inoculating enough virus particles in at least some squash plants to kickstart the infection resulting in widespread epidemics. The third, CuLCrV infection in squash is assisted by mixed infection with CYSDV and/or CCYV. More research is warranted to explore these scenarios to develop long-term areawide management of CuLCrV.

## Figures and Tables

**Figure 1 pathogens-12-01119-f001:**
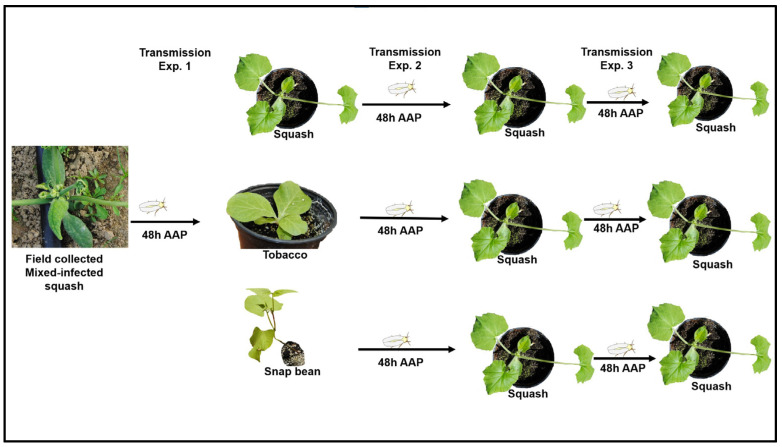
***Bemisia tabaci* B mediated CuLCrV transmission experiments’ sketch.** Field-collected, mixed-infected squash (CuLCrV and CYSDV) plants were used as an initial inoculum source. Transmission experiment one (E1): CuLCrV transmission from mixed-infected squash to tobacco, snap bean, and squash. Transmission experiment two (E2): CuLCrV transmission from mixed- (squash, CuLCrV and CYSDV) versus singly infected (tobacco and snap bean, CuLCrV) plants to squash. Transmission experiment three (E3): (iii), CuLCrV transmission from mixed- versus singly infected squash to squash. In all transmission studies, whiteflies were given a 48 h acquisition access period.

**Figure 2 pathogens-12-01119-f002:**
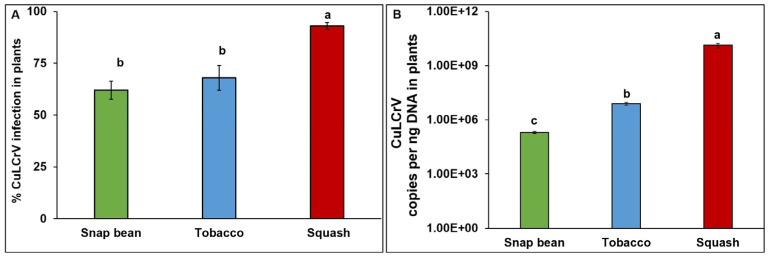
***Bemisia tabaci* B-mediated CuLCrV transmission from field-collected mixed- (CuLCrV and CYSDV: squash) infected inoculum source to squash, tobacco, and snap bean.** (**A**), Percent CuLCrV infection in snap bean, tobacco, and squash plants inoculated from mixed-infected (CuLCrV and CYSDV)—squash plants. (**B**), CuLCrV accumulation in snap bean, tobacco, and squash plants. Values are means  ±  SE. Means with different letters are significantly different (HSD test at *p*  <  0.05). Different lowercase letters on bars indicate differences between treatments.

**Figure 3 pathogens-12-01119-f003:**
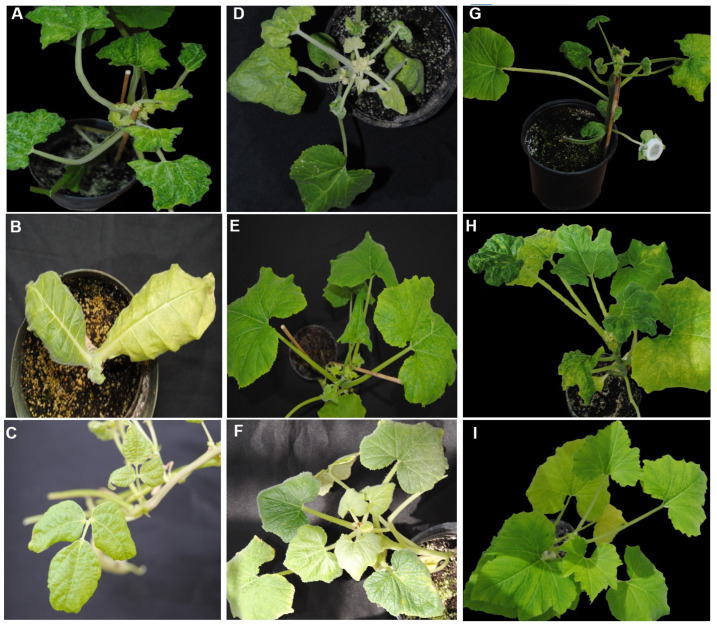
**Symptom development in plants infected with CuLCrV and/or CYSDV.** Photographs were taken at the most symptomatic phase of infection in respective plants. (**A**), mixed-infected squash with CuLCrV and CYSDV; (**B**), tobacco infected with CuLCrV; (**C**), snap bean infected with CuLCrV; (**D**) CuLCrV- and CYSDV-infected squash inoculated by whiteflies that acquired CuLCrV and CYSDV from mixed-infected squash; (**E**), CuLCrV-infected squash inoculated by whiteflies that acquired virus from singly infected tobacco; (**F**), CuLCrV-infected squash inoculated by whiteflies that acquired virus from singly infected snap bean; (**G**), CuLCrV- and CYSDV-infected squash inoculated by whiteflies that acquired CuLCrV and CYSDV from mixed-infected squash that was inoculated by whiteflies that acquired both viruses from mixed-infected squash; (**H**), CuLCrV-infected squash inoculated by whiteflies that acquired CuLCrV from squash inoculated by whiteflies post-acquisition on singly infected tobacco; (**I**), CuLCrV-infected squash inoculated by whiteflies that acquired CuLCrV from squash inoculated by whiteflies post-acquisition on singly infected snap bean plants.

**Figure 4 pathogens-12-01119-f004:**
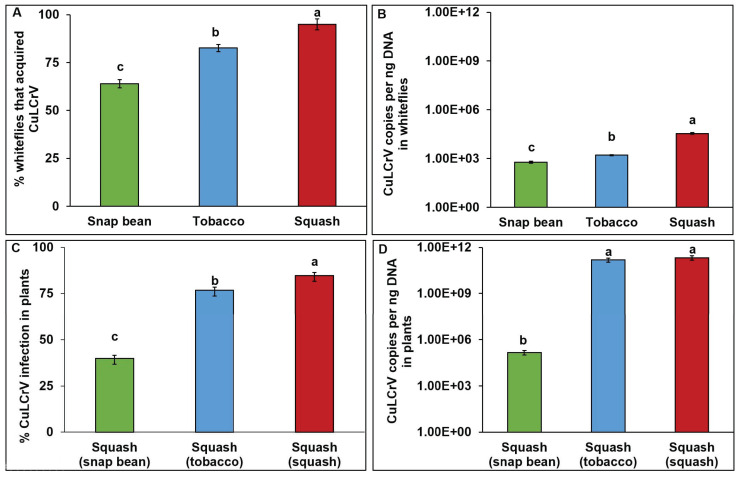
***Bemisia tabaci* B-mediated CuLCrV transmission from singly (CuLCrV: tobacco and snap bean)- and mixed (CuLCrV and CYSDV: squash)- infected inoculum sources to squash.** (**A**), Percentage of whiteflies that acquired CuLCrV by feeding on singly (CuLCrV: tobacco and snap bean)- or mixed-infected (CuLCrV and CYSDV: squash) plants. (**B**), CuLCrV accumulation in whiteflies that acquired CuLCrV by feeding on singly (CuLCrV: tobacco and snap bean)- or mixed-infected (CuLCrV and CYSDV: squash) plants. (**C**), Percent infection in squash inoculated by whiteflies post-acquisition on singly (CuLCrV: tobacco and snap bean)- or mixed-infected (CuLCrV and CYSDV: squash) plants. (**D**), CuLCrV accumulation in squash inoculated by whiteflies post-acquisition on singly (CuLCrV: tobacco and snap bean)- or mixed-infected (CuLCrV and CYSDV: squash) plants. Values are means  ±  SE. Means with different letters are significantly different (*p*  <  0.05).

**Figure 5 pathogens-12-01119-f005:**
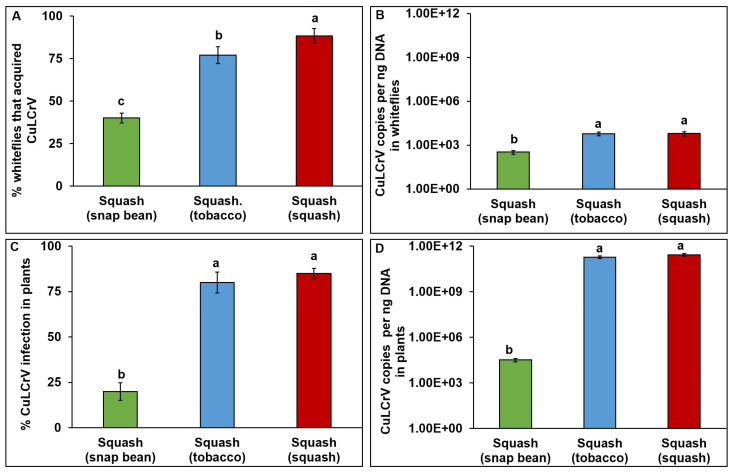
***Bemisia tabaci* B-mediated CuLCrV transmission from singly infected (squash inoculated by whiteflies post-acquisition on singly infected snap bean or tobacco plants) and mixed-infected squash plants (squash inoculated by whiteflies post-acquisition on mixed infected squash) to squash plants.** (**A**), Percentage of whiteflies that acquired CuLCrV feeding on singly infected (CuLCrV: squash inoculated by whiteflies post-acquisition on bean or tobacco) or mixed-infected (CuLCrV and CYSDV: squash plants inoculated by whiteflies post-acquisition on mixed-infected squash) plants. (**B**), CuLCrV accumulation in whiteflies feeding on singly infected (CuLCrV: squash inoculated by whiteflies post-acquisition on snap bean or tobacco) or mixed-infected (CuLCrV and CYSDV: squash (squash inoculated by whiteflies post-acquisition on mixed-infected squash)) plants. (**C**), Percent CuLCrV infection in squash plants inoculated from singly infected (squash inoculated by whiteflies post-acquisition on infected bean or tobacco) or mixed-infected (CuLCrV and CYSDV) squash (squash inoculated by whiteflies post-acquisition on mixed-infected squash) plants. (**D**), CuLCrV accumulation in squash inoculated from singly infected (CuLCrV: squash inoculated by whiteflies post-acquisition on infected snap bean or tobacco) or mixed-infected (CuLCrV and CYSDV: squash inoculated by whiteflies post-acquisition on mixed infected squash) plants. Means with different letters are significantly different (*p*  <  0.05) (please check once more).

## Data Availability

Not applicable.

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
