# Peer review of "Effects of Host Plants and Their Infection Status on Acquisition and Inoculation of A Plant Virus by Its Hemipteran Vector"

_pathogens, 2023, doi:10.3390/pathogens12091119_

Round 1
Reviewer 1 Report
The manuscript of Gautam et al. describes diferences of begomovirus and crinivirus in single and mixed infections in squash, snap bean, and tobacco. Diferences in virus titers were observed in diferent plants and in viral acquisition by whiteflies. One of the main suggestions is in the introduction section, since only reports of the cited viruses and experimental assays were described. I suggest to discuss about the importance of CuLCrV, CYDSV, and CCYV on squash, snap bean, and tobacco or other hosts (e.g.; incidence, damage). How importante is CuLCrV on tobacco? I suggest to use the term “MEAM1 species” insted “B biotype” for whitefly description in the whole manuscript. I also suggest to do a comparison in the discussion section on before and after the introduction of the whitefly MEAM1 species in the USA, which may support the increased importance of begomoviruses such as CuLCrV (see L442-444). Based on my comments as outlined below, I recommend minor revisions.
Abstract
L21: The whitefly, Bemisia tabaci (Gennadius) (Hemiptera: Aleyrodidae) Middle East-Asia Minor 1 (MEAM1) species transmits ...
Introduction
L53: I suggest to use the term MEAM1 instead B biotype.
L52-56: The authors mentioned that the CuLCrV has increasingly important in squash and snap bean, however no reference was provided. Please, explain.
53-54: ... transmitted cucurbit leaf crumple virus (CuLCrV) (family Geminiviridae, genus Begomovirus), member of the species Cucurbit leaf crumple virus, ...
L60: ... studied previously (reference). However, the ability ...
L84: Cucurbitacea (italic)
Materials and Methods
Why was used tobacco?
L151: How many whiteflies were attached per plant?
L156-158: move this sentence to the results section.
L180-181: How many whiteflies were tested per each assay? Was measured virus accumulation for a single insect?
Results
L210: Where the results of CYSDV transmission assays? Please, describe.
Discussion
L394: Please, describe the exact IAP. 24? 48?
Author Response
August 23, 2023
Dear Reviewer,
My co-authors and I greatly appreciate your comments. We believe that your review and comments have improved our manuscript substantially. We have carefully considered each comment and have tried to address them to the best of our abilities. Our explanation for each comment is included below. Our replies are in bold font. The revisions also are tracked on the manuscript with a red font.
Reviewer 1
Comments and Suggestions for Authors
The manuscript of Gautam et al. describes diferences of begomovirus and crinivirus in single and mixed infections in squash, snap bean, and tobacco. Diferences in virus titers were observed in diferent plants and in viral acquisition by whiteflies. One of the main suggestions is in the introduction section, since only reports of the cited viruses and experimental assays were described. I suggest to discuss about the importance of CuLCrV, CYDSV, and CCYV on squash, snap bean, and tobacco or other hosts (e.g.; incidence, damage). How importante is CuLCrV on tobacco? I suggest to use the term “MEAM1 species” insted “B biotype” for whitefly description in the whole manuscript. I also suggest to do a comparison in the discussion section on before and after the introduction of the whitefly MEAM1 species in the USA, which may support the increased importance of begomoviruses such as CuLCrV (see L442-444). Based on my comments as outlined below, I recommend minor revisions.
Abstract
L21: The whitefly, Bemisia tabaci (Gennadius) (Hemiptera: Aleyrodidae) Middle East-Asia Minor 1 (MEAM1) species transmits ...
We have used Bemisia tabaci B on the manuscript to generically reflect the newer B. tabaci taxonomy publications that focused on whole genome sequencing as well as mitochondrial COI sequencing (de Moya et al.2019; Paredes‐Montero et al. 2020). In these publications, both B and Q cryptic species seem to share a single biogeographical lineage referred to as the NF-MED-ME (de Moya et al.2019; Paredes‐Montero et al. 2020) and are not represented as MEAM1 and MED lineages. As you are aware, Bemisia tabaci cryptic species nomenclature has had numerous iterations and remains ambiguous.
In that context, we completely agree with you. Nomenclature needs to be at least clarified on the manuscript. For that reason, as you suggested, we have used the following description at first mention: Bemisia tabaci B (otherwise referred to as B biotype/B mitotype/MEAM1). We have added two additional references as well. We believe that this cross reference should resolve any confusion among readers.
de Moya et al.2019 Diversity 2019, 11(9), 151; https://doi.org/10.3390/d11090151;
Paredes‐Montero et al. 2020 https://esajournals.onlinelibrary.wiley.com/doi/full/10.1002/ecs2.3154
Introduction
L53: I suggest to use the term MEAM1 instead B biotype.
An explanation is included above for usage of B cryptic species instead of MEAM1. Currently in Line 53.
L52-56: The authors mentioned that the CuLCrV has increasingly important in squash and snap bean, however no reference was provided. Please, explain.
A reference has been added. Line 57.
53-54: ... transmitted cucurbit leaf crumple virus (CuLCrV) (family Geminiviridae, genus Begomovirus), member of the species Cucurbit leaf crumple virus, ...
Edited as suggested. Line 55.
L60: ... studied previously (reference). However, the ability ...
A reference has been added. Line 62.
L84: Cucurbitacea (italic)
Italicized as suggested. Line 86.
Materials and Methods
Why was used tobacco?
A couple of reasons why we considered using tobacco.
It was initially used to assess if it can be used as a host to see if we could establish single infection of CuLCrV. Tobacco was then included as an experimental host and is mentioned on the manuscript.
Tobacco is a field crop in South GA and North Florida. Approximately 20,000 acres are grown here. It is not clear whether CuLCrV could infect tobacco. Although this needs to be studied. The implications of tobacco influencing virus epidemics is minimal in the current cropping system.
Tobacco was included as an experimental host and is mentioned on the manuscript. Line 102.
L151: How many whiteflies were attached per plant?
~100. This information is included in line 158.
L156-158: move this sentence to the results section.
Modified as suggested. Currently in line 229.
L180-181: How many whiteflies were tested per each assay? Was measured virus accumulation for a single insect?
Yes, it was individual whiteflies. This information is included in lines 187-189.
Results
L210: Where the results of CYSDV transmission assays? Please, describe.
Since this manuscript exclusively discusses CuLCrV acquisition and transmission, CYSDV information was neither gathered nor presented as part of this manuscript. However, in another study, we studied single and mixed infection of CuLCrV and CYSDV and their interactions in plants as well as in whiteflies, and that information is already published (Gautam et al. 2020). Overall, there seems to be antagonistic effect when present together leading to reduction in CYSDV levels in mixed-infected plants as opposed to plants infected with CYSDV alone.
Gautam, S., K. R. Gadhave, J. W. Buck, B. Dutta, T. Coolong, S. Adkins, and R. Srinivasan. 2020. Virus-virus interactions in a plant host and in a hemipteran vector: Implications for vector fitness and virus epidemics. Virus Research. 286: 198069. Doi: 10.1016/j.virusres.2020.198069.
Discussion
L394: Please, describe the exact IAP. 24? 48?
The IAP used in the Wintermantel Study was 48h. As suggested by the other reviewer sentence was removed from the revised version.
Yours sincerely,
Rajagopalbabu Srinivasan

Reviewer 2 Report
This paper describes the dependence of the virus concentration in the acquisition host on the transmission efficiency of CuLCrV by WFs.
In itself an interesting topic and indeed agriculturally important however the paper does suffer from flaws which should be addressed and corrected before it would be acceptable for publication. I recommend major revision.
One important point is that the authors incorrectly use the term ‘viruliferous’ throughout the paper for WFs that apparently test positive for CuLCrV in PCR (i.e. on pg 7, ln 242). However, technically speaking one cannot use the word ‘viruliferous’ here as the test only shows the ingestion (or presence) of virus in the WFs not their ability to transmit.
In that respect the legends on the Y-axis of Figs 4A and 5A are incorrect and should not read ‘infection in WFs’ but rather something like % of WFs that acquired virus.
On Pg 4 lns 176-177 mentions that WFs were placed on cotton plants. Presumably the WFs were put on the cotton plants to clear their guts of any ingested virus so after 48 hrs the virus measured in the individual WFs can be assumed to be present in the hemolymph and salivary glands (see my remark above on the incorrect use of the term). That may explain the use of the word ‘viruliferous’ by the authors. This is however not explained to the reader. If this indeed the case it should be much better clarified in the text.
Related to this is Pg 11, ln 409. This speculation is not supported by data from this study. No data on actual virus levels in WF salivary glands are presented. One cannot directly assume that total virus levels in the WF body represent virus concentrations in the salivary glands and thus ‘infectivity’.
In addition, this point should also be addressed in the discussion. If indeed the experiments in this paper are based on an intermediate feeding period on a non-host for CuLCrv (cotton), how likely is it that his will also be a natural situation in the field and to what extent will this affect the transmission from hosts with different virus concentrations?
A second point is that this paper is primarily based on the levels of ‘virus accumulation’ as determine by PCR in the different plant species. However, these differences in ‘virus accumulation’ in the different plant species are really small (Pg7, lns 246-250 and ln 266). The authors should therefor supply the ‘raw’ data on virus titers in the different samples and repetitions between experiments in a separate table so readers can compare themselves.
A third point is that the initial squash material was infected with CYSDV and that he authors themselves discuss the possible influence of this co-infection on CuLCrV titers in the source and recipient plants (i.e. pg 9, ln 329). What I miss in general throughout the paper are the concentrations of CYSDV in all the plants. It is clear that the presence of CYSDV has an effect on the accumulation of CuLCrV so why are these concentrations not taken into account? Why do the authors ignore the presence and effect of CYSDV in this discussion (pg 10, ln 368 and 371)?
Please provide this information and discuss.
In Section 3.3 (Ln 279-344) the descriptions of the different plants and experiments in this section (but basically throughout the paper) is very (almost overly) complicated. For example, the different abbreviations in this section for the 3 individual repetitions of Exp 3. The current use of E1-3 is too confusing in comparison with the earlier exp 1 and 2 (E-1 and E2). Please come up with a better to understand scheme. Maybe based on Fig 1 ( S - S - S and T - S - S and B - S - S)?
Other more detailed remarks
Pg3, ln 130: this paragraph does not describe the maintenance of the virus, only the extraction of NA from leaf material. Please describe how initial squash inoculum plants were maintained and how their infection status was confirmed.
Pg 4, ln 148: it is unclear what the inoculum source of the viruses is. Paragraph 2.2 suggest to describe the initial inoculum source of the viruses but only describes the RNA/DNA extraction procedure. It seems unlikely that the surface sterilized leaves were used as inoculum source. Please clarify. Please also provide details on the virus concentrations (both viruses) in the source plants and how their infection status was determined. How many source plants were used for the acquisition experiments? Was only one (and the same) source plant used to inoculate all 3 plant species?
Pg 4, ln 151: please list how may WFs were used per clip cage and per plant in total
Pg 4, line 157: so was this due to the nonhost status of tobacco and snap bean for CYSDV?
Pg 4, ln 159: were CYSDV concentrations determined? If not, why not?
Pg 4, ln 180: was the CuLCrV concentration in the WF compared to the virus concentration in the source plants?
Pg 4, ln 176: please list how many WF were transferred to the cotton plants in total and per clip cage.
4-9
Pg 4, ln 177: how many WF were used to test for virus acquisition by PCR?
Pg 3, ln 159: what is the difference between inoculated and infected plants? Please explain.
Pg 3, ln 168: how do you know plants are mixed infected when only a test for CuLCrV is performed. Please correct
Pg 5, ln 182: how were the non- viruliferous WFs obtained? Were these fed (AAP) on healthy squash?
Pg 5, ln 186: CYSDV accumulation (concentration) was measured so why are these data not presented?
Pg 11, ln 384: this is a logical remark because it merely indicated that amount of virus ingested by the WF. As such is superfluous and can be deleted. Total virus accumulation in the WF is different from accumulation in the salivary glands and should not be regarded as an indicator of the level of efficacy of transmission.
Pg 11, lns 386-396: this paragraph should be deleted as it does not contribute to the paper. Firstly, no data are presented on the levels of CYSDV in the source plants which is a clear omission in this paper (see previous remarks). Secondly because it only stresses the question why the authors had not previously tested their snap bean cv for its susceptibility to CYSDV if they would have thought it would be an factor to consider. Now it reads like an excuse.
Pg 11, lns 400: the authors fail to discuss another explanation for this recovery phenotype. Can they rule out that replication of the virus in snap bean actually selects for a virus variant less capable to replicate in squash and may thus lead to lower virus titre in squash?
Pg 12, ln 446: No, the current study does not show that CuLCrV epidemics largely depend on the numbers of virus particles and transmitted. Even though the authors mention some possible reasons in the sentences to follow, there might be totally different (agricultural ) reasons for epidemics to occur. The authors should delete this conclusion.
Additional textual remarks
Abstract, ln 23: please delete the word ‘other’ as this suggest CuLCrV is also a crinivirus
Abstract, ln 29: plant species ( not plants)
Pg 2, ln 60: please provide reference for these previous studies
Pg 2, ln 91: THE inoculum source ....
Pg 5, ln 194: squash (obtained) ...so move parenthesis
Pg 5, ln 195: remove squash after parenthesis
Pg 5 , lns 210-213. Please combine these two sentences as the contain the same information
Pg 8, ln 282: please correct English; the first, second and third exp. See also ln 320.
Ln 377: lower case for Studies
Ln 381: APP should be AAP
Pg9, lns 309-310: please check position of parenthesises as these are currently incorrect.
Pg 11, ln 410: should this not read ‘second transmission exp’?
Limited English corrections required
Author Response
August 23, 2023
Dear Reviewer,
My co-authors and I greatly appreciate your comments. We believe that your review and comments have greatly improved our manuscript. We have carefully considered each comment and have tried to address them to the best of our abilities. Our explanation for each comment is included below. Our replies are in bold font. The revisions also are tracked on the manuscript with a red font.
Comments and Suggestions for Authors
This paper describes the dependence of the virus concentration in the acquisition host on the transmission efficiency of CuLCrV by WFs.
In itself an interesting topic and indeed agriculturally important however the paper does suffer from flaws which should be addressed and corrected before it would be acceptable for publication. I recommend major revision.
One important point is that the authors incorrectly use the term ‘viruliferous’ throughout the paper for WFs that apparently test positive for CuLCrV in PCR (i.e. on pg 7, ln 242). However, technically speaking one cannot use the word ‘viruliferous’ here as the test only shows the ingestion (or presence) of virus in the WFs not their ability to transmit.
In that respect the legends on the Y-axis of Figs 4A and 5A are incorrect and should not read ‘infection in WFs’ but rather something like % of WFs that acquired virus.
Figs 4A and 5A legends edited as suggested.
On Pg 4 lns 176-177 mentions that WFs were placed on cotton plants. Presumably the WFs were put on the cotton plants to clear their guts of any ingested virus so after 48 hrs the virus measured in the individual WFs can be assumed to be present in the hemolymph and salivary glands (see my remark above on the incorrect use of the term). That may explain the use of the word ‘viruliferous’ by the authors. This is however not explained to the reader. If this indeed the case it should be much better clarified in the text.
The whitefly population used in this study is a very efficient vector of CuLCrV. Upon acquisition, it is anticipated that the virus would translocate to the salivary glands. However, to avoid any confusion, we have now replaced ‘viruliferous’ and ‘non-viruliferous’ as whiteflies that acquired CuLCrV and whiteflies that did not acquire CuLCrV.
Related to this is Pg 11, ln 409. This speculation is not supported by data from this study. No data on actual virus levels in WF salivary glands are presented. One cannot directly assume that total virus levels in the WF body represent virus concentrations in the salivary glands and thus ‘infectivity’.
In addition, this point should also be addressed in the discussion. If indeed the experiments in this paper are based on an intermediate feeding period on a non-host for CuLCrv (cotton), how likely is it that his will also be a natural situation in the field and to what extent will this affect the transmission from hosts with different virus concentrations?
We agree with you. The amount of virus in salivary glands may or may not correlate with what’s present in other tissues/whole body. This point has been explained in the discussion section in line 402.
A second point is that this paper is primarily based on the levels of ‘virus accumulation’ as determine by PCR in the different plant species. However, these differences in ‘virus accumulation’ in the different plant species are really small (Pg7, lns 246-250 and ln 266). The authors should therefor supply the ‘raw’ data on virus titers in the different samples and repetitions between experiments in a separate table so readers can compare themselves.
The qPCR data are now included in the supplementary file as suggested.
A third point is that the initial squash material was infected with CYSDV and that he authors themselves discuss the possible influence of this co-infection on CuLCrV titers in the source and recipient plants (i.e. pg 9, ln 329). What I miss in general throughout the paper are the concentrations of CYSDV in all the plants. It is clear that the presence of CYSDV has an effect on the accumulation of CuLCrV so why are these concentrations not taken into account? Why do the authors ignore the presence and effect of CYSDV in this discussion (pg 10, ln 368 and 371)?
Please provide this information and discuss.
This is a valid question and you raise an important point. Since this manuscript exclusively discusses CuLCrV acquisition and inoculation/transmission, CYSDV information was neither gathered nor presented as part of this manuscript. However, in another study, we exclusively studied single and mixed infection of CuLCrV and CYSDV and their interactions in plants as well as in whiteflies, and that information is already published (Gautam et al. 2020). Overall, there seems to be antagonistic effect when present together leading to reduction in CYSDV levels in mixed-infected plants as opposed to plants infected with CYSDV alone.
Gautam, S., K. R. Gadhave, J. W. Buck, B. Dutta, T. Coolong, S. Adkins, and R. Srinivasan. 2020. Virus-virus interactions in a plant host and in a hemipteran vector: Implications for vector fitness and virus epidemics. Virus Research. 286: 198069. Doi: 10.1016/j.virusres.2020.198069.
In Section 3.3 (Ln 279-344) the descriptions of the different plants and experiments in this section (but basically throughout the paper) is very (almost overly) complicated. For example, the different abbreviations in this section for the 3 individual repetitions of Exp 3. The current use of E1-3 is too confusing in comparison with the earlier exp 1 and 2 (E-1 and E2). Please come up with a better to understand scheme. Maybe based on Fig 1 ( S - S - S and T - S - S and B - S - S)?
We agree with you on confusion in the text generated due to the repeated use of phrases. The text has been edited in the method and results section for clarity.
Other more detailed remarks
Pg3, ln 130: this paragraph does not describe the maintenance of the virus, only the extraction of NA from leaf material. Please describe how initial squash inoculum plants were maintained and how their infection status was confirmed.
Information has been added in line 150. “ The inoculum was maintained in the greenhouse through repeated whitefly-mediated inoculation in whitefly-proof cages by placing two-week-old non-infected squash plants with the infected plants.”
Pg 4, ln 148: it is unclear what the inoculum source of the viruses is. Paragraph 2.2 suggest to describe the initial inoculum source of the viruses but only describes the RNA/DNA extraction procedure. It seems unlikely that the surface sterilized leaves were used as inoculum source. Please clarify. Please also provide details on the virus concentrations (both viruses) in the source plants and how their infection status was determined. How many source plants were used for the acquisition experiments? Was only one (and the same) source plant used to inoculate all 3 plant species?
Field-collected mixed-(CuLCrV and CYSDV) infected squash served as the inoculum source and squash, tobacco, and snap bean as recipients Around one thousand non-viruliferous whiteflies (up to 48h old) feeding on cotton were given a 48h acquisition access period (AAP) on mixed-infected squash. Inoculum source information has been included in lines (136-137; 177-179; 203-206).
The virus concentrations (both viruses) in the source plants and how their infection status was determined have already been published (Gautam et al. 2020).
Gautam, S., K. R. Gadhave, J. W. Buck, B. Dutta, T. Coolong, S. Adkins, and R. Srinivasan. 2020. Virus-virus interactions in a plant host and in a hemipteran vector: Implications for vector fitness and virus epidemics. Virus Research. 286: 198069. Doi: 10.1016/j.virusres.2020.198069.
We collected 15 mixed-infected plants from the field, out of which 3 where used for E1 transmission experiments. This information has been included in lines. Line 137-139; 167-168.
Pg 4, ln 151: please list how may WFs were used per clip cage and per plant in total
About 100 whiteflies were used to inoculate a plant. Currently in lines 158.
Pg 4, line 157: so was this due to the nonhost status of tobacco and snap bean for CYSDV?
Yes, that is correct. Tobacco seems to be a non-host for our isolate CYSDV. According to another reviewer that sentence has now been moved to the results section. Currently in lines 229-230.
Pg 4, ln 159: were CYSDV concentrations determined? If not, why not?
This is a valid question and you raise an important point. Since this manuscript exclusively discusses CuLCrV acquisition and inoculation/transmission, CYSDV information was neither gathered nor presented as part of this manuscript. However, in another study, we exclusively studied single and mixed infection of CuLCrV and CYSDV and their interactions in plants as well as in whiteflies. That information is already published (Gautam et al. 2020). Overall, there seems to be antagonistic effect when present together leading to reduction in CYSDV levels in mixed-infected plants as opposed to plants infected with CYSDV alone.
Pg 4, ln 180: was the CuLCrV concentration in the WF compared to the virus concentration in the source plants?
CuLCrV loads were estimated in the inoculum source as well as in whiteflies post acquisition. All data are presented in figures 4B and 5B.
Pg 4, ln 176: please list how many WF were transferred to the cotton plants in total and per clip cage.
Whiteflies with or without viruses were obtained by allowing whiteflies to feed on non-infected or singly and mixed-infected inoculum sources for a 48h AAP. Following AAP, whiteflies were attached to cotton plants using clip cages (100/clip cage). After 48 h on cotton, twenty whiteflies were randomly collected, and surface sterilized using the six-step protocol. This information has been included in lines 186-188.
4-9Pg 4, ln 177: how many WF were used to test for virus acquisition by PCR?
They were individual whiteflies. This information has now been clarified in the text in lines 188-190.
Pg 3, ln 159: what is the difference between inoculated and infected plants? Please explain.
This has now been modified as infected plants. Line 169.
Pg 3, ln 168: how do you know plants are mixed infected when only a test for CuLCrV is performed. Please correct
That information is included in lines 164-166. The infection status was obtained via endpoint PCR.
Pg 5, ln 182: how were the non- viruliferous WFs obtained? Were these fed (AAP) on healthy squash?
Yes. non-viruliferous whiteflies were given 48h AAP on healthy squash. That information is included in line 187.
Pg 5, ln 186: CYSDV accumulation (concentration) was measured so why are these data not presented?
CYSDV infection status was determined, but the concentrations were not determined mainly for the reasons already explained.
Pg 11, ln 384: this is a logical remark because it merely indicated that amount of virus ingested by the WF. As such is superfluous and can be deleted. Total virus accumulation in the WF is different from accumulation in the salivary glands and should not be regarded as an indicator of the level of efficacy of transmission.
The sentence was removed as suggested.
Pg 11, lns 386-396: this paragraph should be deleted as it does not contribute to the paper. Firstly, no data are presented on the levels of CYSDV in the source plants which is a clear omission in this paper (see previous remarks). Secondly because it only stresses the question why the authors had not previously tested their snap bean cv for its susceptibility to CYSDV if they would have thought it would be an factor to consider. Now it reads like an excuse.
Deleted as suggested.
Pg 11, lns 400: the authors fail to discuss another explanation for this recovery phenotype. Can they rule out that replication of the virus in snap bean actually selects for a virus variant less capable to replicate in squash and may thus lead to lower virus titre in squash?
Information has been added as suggested. Line 387.
Pg 12, ln 446: No, the current study does not show that CuLCrV epidemics largely depend on the numbers of virus particles and transmitted. Even though the authors mention some possible reasons in the sentences to follow, there might be totally different (agricultural ) reasons for epidemics to occur. The authors should delete this conclusion.
Deleted as suggested.
Additional textual remarks
Abstract, ln 23: please delete the word ‘other’ as this suggest CuLCrV is also a crinivirus
Deleted as suggested.
Abstract, ln 29: plant species (not plants)
Edited as suggested. Line 29.
Pg 2, ln 60: please provide reference for these previous studies
Reference added. Line 62.
Pg 2, ln 91: THE inoculum source ....
Edited as suggested. Line 95.
Pg 5, ln 194: squash (obtained) ...so move parenthesis
Edited for clarity.
Pg 5, ln 195: remove squash after parenthesis
Edited for clarity.
Pg 5 , lns 210-213. Please combine these two sentences as the contain the same information
Edited as suggested. Line 231-232.
Pg 8, ln 282: please correct English; the first, second and third exp. See also ln 320.
Edited for clarity as suggested by another reviewer.
Ln 377: lower case for Studies
Edited as suggested. Line 378.
All suggestions have been incorporated (Please check).
Ln 381: APP should be AAP
Corrected. Line 382.
Pg9, lns 309-310: please check position of parenthesises as these are currently incorrect.
Has been edited.
Pg 11, ln 410: should this not read ‘second transmission exp’?
This is the third experiment. Correct as it is. Line 384.
Yours sincerely,
Rajagopalbabu Srinivasan
(Professor)

Reviewer 3 Report
This is a valuable study that addresses important questions about the dynamics of whitefly-borne viruses. The work is described in great detail and looks very thorough and well done. However, the manuscript is difficult to read due to the complexity of the mixed vs. single-infection variables and the various plant hosts. To a non-expert in whitefly viruses such as myself, the important outcomes tend to get lost in massive amounts of wordy detail which becomes repetitive. My overall comment is to ask the authors to seek every way possible to simplify the presentation of results.
Specific comments:
l. 151 describe the "clip cage" and how it works
l. 155. what exactly is a replicate in these experiments? are separate leaves on the same plant counted as replicates, or are they separate plants? Hopefully the latter.
l. 283-285 Each experiment apparently provided infected plants to the following experiment, in sequence. E1 to E2, E2 to E3. Clarify in Methods.
l 323-328. This is a good example of hardly intelligible text, which is common in many parts of the ms, especially in Results.
The first part of the Discussion is very helpful and clear, and I suggest using it as a model for simplifying other parts of the discussion.
With some improvements in clarity and reductions in wordy text length, this paper will be a valuable contribution.
Author Response
August 23, 2023
Dear Reviewer,
My co-authors and I greatly appreciate your comments. We believe that your review and comments have improved our manuscript conisderably. We have carefully considered each comment and have tried to address them to the best of our abilities. Our explanation for each comment is included below. Our replies are in bold font. The revisions also are tracked on the manuscript with a red font.
This is a valuable study that addresses important questions about the dynamics of whitefly-borne viruses. The work is described in great detail and looks very thorough and well done. However, the manuscript is difficult to read due to the complexity of the mixed vs. single-infection variables and the various plant hosts. To a non-expert in whitefly viruses such as myself, the important outcomes tend to get lost in massive amounts of wordy detail which becomes repetitive. My overall comment is to ask the authors to seek every way possible to simplify the presentation of results.
We have edited the text in the results section for clarity.
Specific comments:
- 151 describe the "clip cage" and how it works
We used clip cages made of foam. Each cage comprises two hollow foam rings (inner diameter of 2.5 cm, and a height of 1.5 cm) with mesh covering one side of each ring for visibility and ventilation. As shown in the supplementary figure 1, using pins plant leaf tissues were sandwiched between the rings of clip cages. This information is included in the methods section in lines 160.
- 155. what exactly is a replicate in these experiments? are separate leaves on the same plant counted as replicates, or are they separate plants? Hopefully the latter.
Yes, that is correct. Individual plants were considered as replicates in each experiment. The experiment was then repeated in time. Lines 166-169.
- 283-285 Each experiment apparently provided infected plants to the following experiment, in sequence. E1 to E2, E2 to E3. Clarify in Methods.
The inoculum source for each experiment has been explained in the method section. Also, the text has been edited in the method and result section for clarity. E1: Line 167; E2:177; E3: 203.
l 323-328. This is a good example of hardly intelligible text, which is common in many parts of the ms, especially in Results.
The first part of the Discussion is very helpful and clear, and I suggest using it as a model for simplifying other parts of the discussion.
With some improvements in clarity and reductions in wordy text length, this paper will be a valuable contribution.
The text has been edited in the method and result sections for clarity as suggested, and the changes are highlighted in red in the revised version.
Yours sincerely,
Rajagopalbabu Srinivasan
(Professor)

Round 2
Reviewer 2 Report
The authors have dealt adequately with most of the initial comments. I do think that not taking the CYSDV concentrations into account is an experimental flaw that should be taken into consideration future studies (if relevant)